# Social and Behavioural Correlates of High Physical Activity Levels among Aboriginal Adolescent Participants of the Next Generation: Youth Wellbeing Study

**DOI:** 10.3390/ijerph20043738

**Published:** 2023-02-20

**Authors:** Rona Macniven, Christopher D. McKay, Simon Graham, Lina Gubhaju, Robyn Williams, Anna Williamson, Grace Joshy, John Robert Evans, Robert Roseby, Bobby Porykali, Aryati Yashadhana, Rebecca Ivers, Sandra Eades

**Affiliations:** 1School of Population Health, Faculty of Medicine & Health, UNSW Sydney, Sydney, NSW 2052, Australia; 2Melbourne School of Population and Global Health, The University of Melbourne, Melbourne, VIC 3010, Australia; 3Department of Infectious Diseases, Peter Doherty Institute for Infection and Immunity, The University of Melbourne, Melbourne, VIC 3010, Australia; 4Curtin Medical School, Curtin University, Perth, WA 6102, Australia; 5National Centre for Epidemiology and Population Health, College of Health & Medicine, Australian National University, Canberra, ACT 2601, Australia; 6Moondani Toombadool Centre, Swinburne University of Technology, Melbourne, VIC 3122, Australia; 7Department of Respiratory Medicine, Monash Children’s Hospital, Clayton, VIC 3168, Australia; 8Department of Paediatrics, School of Clinical Sciences, Monash University, Melbourne, VIC 3800, Australia; 9Guuna-Maana (Heal) Aboriginal and Torres Strait Islander Health Program, The George Institute for Global Heath, Sydney, NSW 2042, Australia; 10Centre for Primary Health Care & Equity, UNSW Sydney, Sydney, NSW 2052, Australia

**Keywords:** healthy lifestyle, exercise, screen time, peer influence, health priorities, demography

## Abstract

Physical activity typically decreases during teenage years and has been identified as a health priority by Aboriginal adolescents. We examined associations between physical activity levels and sociodemographic, movement and health variables in the Aboriginal led ‘Next Generation: Youth Well-being (NextGen) Study’ of Aboriginal people aged 10–24 years from Central Australia, Western Australia and New South Wales. Baseline survey data collected by Aboriginal researchers and Aboriginal youth peer recruiters from 2018 to 2020 examined demographics and health-related behaviours. Logistic regression was used to estimate odds ratios (OR) for engaging in high levels of physical activity in the past week (3–7 days; 0–2 days (ref), or ‘don’t remember’) associated with demographic and behavioural factors. Of 1170 adolescents, 524 (41.9%) had high levels of physical activity; 455 (36.4%) had low levels; 191 (15.3%) did not remember. Factors independently associated with higher odds of physical activity 3–7 days/week were low weekday recreational screen time [55.3% vs. 44.0%, OR 1.79 (1.16–2.76)], having non-smoking friends [50.4% vs. 25.0%, OR 2.27 (1.03–5.00)] and having fewer friends that drink alcohol [48.1% vs. 35.2%, OR 2.08 (1.05–4.14)]. Lower odds of high physical activity were independently associated with being female [40.2% vs. 50.9%, OR 0.57 (0.40–0.80)] and some findings differed by sex. The NextGen study provides evidence to inform the co-design and implementation of strategies to increase Aboriginal adolescent physical activity such as focusing on peer influences and co-occurring behaviours such as screen time.

## 1. Introduction

Aboriginal peoples are the original inhabitants and custodians of Australia with great cultural and linguistic diversity. One-third of Aboriginal people are aged 10–24 years [1], the period typically defined as adolescence and young adulthood [2,3]. Adolescence is often a period of significant change and an opportunity to develop healthy lifestyle behaviours that may carry into adulthood and contribute to good health [2]. Physical activity involves body movement that increases energy expenditure, includes different types and contexts such as exercise and sport participation as well as varying frequency, duration and intensity, and has health and wellbeing benefits for adolescents and young people [4,5]. Physical activity has been identified as a priority for holistic health and wellbeing by Aboriginal adolescents and young people [6]. Evidence from studies with Aboriginal participants demonstrates the benefits of physical activity on key metabolic and health indicators such as reductions in weight and improvement in biomarkers including blood pressure and cholesterol [7].

Between the ages of 10 and 24 years is a critical time to engage Aboriginal people in physical activity [2]. National surveillance and previous studies have highlighted that there are high levels of physical activity in childhood [8,9,10] but these levels decrease during teenage years [11]. Furthermore, several studies indicate that Aboriginal adolescent females are less likely to participate in physical activity compared to Aboriginal adolescent males [8,10,11,12].

Physical activity has often been examined in the context of sedentary and sleep behaviours and need to be considered in combination [13,14]. Integrated movement guidelines acknowledge that relationships exist among individual behaviours of physical activity, sedentary behaviour and sleep [14]. Evidence is emerging of movement behaviour associations among Aboriginal children. Among children aged 5–9 years, low sport participation clustered with low sleep duration and high sport participation clustered with low screen time [15]. In the Longitudinal Study of Indigenous Children (LSIC), high levels of physical activity among children aged 8–13 years children were associated with low levels of playing electronic games and total screen time [16].

Physical activity has also been examined with smoking and alcohol use, as these health behaviours often group together and can predict health and wellbeing outcomes [17]. Further, adolescent and young adult smoking and alcohol behaviours are influenced by their peers, as well as by family, socioeconomic and cultural factors [18,19]. There is evidence that health behaviours group together among Aboriginal adults where being physically active may group with non-smoking and low or no alcohol consumption [20,21]. Strategies that address multiple behaviours simultaneously are likely to be acceptable within community-controlled settings that understand the social and cultural determinants of Aboriginal health [20].

However, after the age of 13 years, very little is known about movement or other health behaviours associated with physical activity. Since this is a critical time when physical activity levels start to decline [11], the current study was undertaken to fill this gap in knowledge. Using data from The ‘Next Generation: Youth Well-being Study’ (NextGen), an Aboriginal-led cohort study of 1200 Aboriginal adolescents and young people aged 10–24 years from Central Australia (CA), Western Australia (WA) and New South Wales (NSW) [3], we examined the sociodemographic, movement and health behaviours associated with physical activity.

## 2. Materials and Methods

This cross-sectional study used data from the NextGen study; recruitment and study methods have been described in detail previously [3].

### 2.1. Participants

Study participants were from the ‘Next Generation: Youth Well-being study’ (NextGen) [3]. Participants were eligible if aged between 10 and 24 years, self-identified as Aboriginal and/or Torres Strait Islander, resided in Western Australia (WA), New South Wales (NSW), or Central Australia (CA), and were able to read and respond to survey questions in English. From March 2018 to March 2020, participants were recruited by teams of Aboriginal research officers via community and peer networks (including Aboriginal community centres, sporting clubs and youth centres), researcher networks, secondary schools and social media. Participants were recruited from a mix of urban, regional and remote areas and were invited to complete a health and wellbeing survey and a clinical assessment of cardio-metabolic risk factors. Of 1309 total recruited, 1292 participants were included in this study. Exclusion reasons included being pregnant, having an eating disorder or a congenital condition (n = 17).

Partnerships and relationships were established with Aboriginal community organisations at all study sites. Survey development was informed by a qualitative exploration of the meaning of health and well-being for adolescents and young people, through formative focus groups and interviews with Aboriginal young people, parents/carers and healthcare providers. Data were collected by Aboriginal researchers and Aboriginal youth peer recruiters, who were essential for engaging young people.

### 2.2. Measures

Participants self-completed surveys on REDCap (research electronic data capture) on tablets or paper-based versions. The surveys covered different topics that included the social determinants of health; Aboriginal cultural engagement/family and community connections; physical health and injury; and social and emotional well-being. Survey questions were identified by investigators and sourced, where possible, from existing studies with Aboriginal people that used reliable and valid measures, including the WA Aboriginal Child Health Survey, the NSW Child Population Health Survey [22] and the Study of Environment on Aboriginal Resilience and Child Health [23].

The main outcome variable was derived from the question “outside of school hours, in the past week, how many days did you exercise or play sport or games that made you sweat and breath hard (e.g., basketball, netball, football, riding a bike, running?” [24] Responses were 0, 1, 2, 3, 4, 5, 6, 7 days or ‘I don’t remember’. We defined 0–2 days as ‘low’ and 3–7 days as ‘high’, consistent with a previous study [12] and aligned to physical activity guidelines [25]. Twelve sociodemographic, other movement and health behaviour variables were used as covariates in this study, given their established associations or relationships with physical activity participation [13,17].

Sociodemographic variables were sex (male, female) and age which was categorised by combining two to three consecutive ages together to retain sufficient category numbers (10–11 years; 12–13 years; 14–15 years; 16–17 years; 18–19 years; 20–21 years; 22–24 years). A sub-sample of participants also provided their home postcode and this was used to determine Indigenous Relative Socioeconomic Outcomes Index (IRSEO) tertiles (most advantaged; middle-advantaged; most disadvantaged [26]). Other movement and health behaviour co-variables were screen time, sleep quality, smoking and alcohol. Screen time was measured using responses to the questions ‘how many hours a day do you usually spend watching TV and using the internet to play games, stream tv shows/movies or use social media’, separately for weekdays and weekends (low 0–2 h/day; high ≥ 3 h/day; don’t remember’) [16]. Sleep quality was measured through the question ‘During the past month, how good has your sleep been at night?’ (very good, fairly good, fairly/very bad) [27]. Participants were asked if they had ever tried smoking a tobacco cigarette (no; yes; prefer not to say). They were also asked ‘how many of your friends smoke cigarettes? [28] (none; a few; about half; all or most). They were also asked if they had ever had a full serving of alcohol (no; yes) [28] and ‘how many of your friends drink alcohol? (none; don’t know; a few; about half; all or most).

### 2.3. Data Analysis

A strengths-based approach was adopted in the analysis, with a focus on associations between positive rather than negative health behaviour categories [29]. The number and percentage of adolescents and young people achieving high levels of physical activity across sociodemographic factors, other movement and health behaviour variables were calculated. Chi-squared tests were used to examine unadjusted association between each explanatory variable and physical activity levels (High physical activity; yes, no). Variables that showed significant association in the unadjusted tests were selected for further investigation.

Multinomial logistic regression models were used to examine the associations between high physical activity (defined as 3–7 days per week) and sociodemographic factors, as well as other indicators of movement and health behaviour that showed significant association with and achieving high levels of physical activity in the unadjusted tests. Analyses were conducted for the whole sample and stratified by sex and were adjusted for each variable within the model. These comprised the following: age (ref: “10–15 y”); Weekday recreational screen time (ref low 0–2 h/day); Weekend recreational screen time (ref low 0–2 h/day); Sleep quality (ref fairly/very bad); Ever tried smoking (ref yes); How many friends smoke (ref all of most); Ever tried a full serve of alcohol (ref yes); How many friends drink alcohol (ref none).

Subsequently, a series of multinomial logistic regression models examined the associations between each variable that was selected for further investigation in the unadjusted tests and achieving higher physical activity. The series of models were conducted among the whole sample and stratified by sex and adjusted for age categories and site. A complete case analysis approach was used for all models.

Adjusted odds ratios (ORs) and their 95% confidence interval (CI) were reported, noting that these do not approximate risk ratios. Statistical analyses were conducted using IBM SPSS statistical software (version 26.0; SPSS Inc., Chicago, IL, USA). For all statistical tests, a *p* value of <0.05 was used to indicate statistical significance.

## 3. Results

A total of 1170 NextGen participants provided a valid response to the question on ‘physical activity days’ and were included in this complete case analysis; physical activity data was missing for 74 participants (5.9%). There were significant variations between participants with complete and missing physical activity data for age (3.2% for 10–15 yr; 9.8% for 16–24 yr), site (4.5% for CA, 2.5% for NSW, 7.7% for WA) but not for sex or IRSEO. Forty-two percent (n = 524) responded to participating in high levels (3+ days) of physical activity, 36% (n = 455) had low levels of physical activity and 15% (n = 191) selected ‘don’t remember’. In the chi-squared analyses, there were significant variation in physical activity for 10 of the 11 sociodemographic and other movement and health behaviour variables (Table 1). Just over half (51%) of males had high levels of physical activity, compared to 40% of females. Fifty-five percent of 10–11 year olds had high levels of physical activity, compared to 27% of 16–24 year olds. In CA and NSW, 41% of adolescents and young people had high levels of physical activity, and 47% in WA.

Figure 1 highlights an overall linear decline in achieving high levels of physical activity as age increases among the whole sample and both males and females, with lower proportions of females achieving high levels of physical activity than males at all ages other than at age 10 years.

In the fully adjusted model (Table 2) and among all participants, females had lower odds of participating in high levels of physical activity than males (40.2% versus 50.9%: OR = 0.57; 95% CI, 0.41–0.81). Among males, those aged 10–11 years, 12–13 years and 18–19 years had higher odds of participating in high levels of physical activity than males aged 22–24 years (54.1% versus 38.9%: OR = 6.83; 95% CI 1.43–35.57; 59.3% versus 38.9%: OR = 4.80; 95% CI, 1.10–20.99; and 51.2% versus 38.9%: OR = 4.59; 95% CI, 1.06–19.96, respectively). Compared to participants with high weekday recreational screen time, those with low weekday screen time had 1.7 times high odds of achieving high physical activity levels (55.3% versus 44.0%: OR = 1.70; 95% CI, 1.11–2.61) among all participants and 2.2 times high odds (62.4% versus 46.9%: OR = 2.20; 95% CI, 1.12–4.30) among males. Compared to participants who reported that all or most of their friends drink alcohol, those who reported that about half of their friends drank had 2.1 times higher odds of achieving high physical activity levels (48.1% versus 35.2%: OR = 2.08; 95% CI, 1.05–4.14) among all participants. Females who had never tried smoking had 2.4 times higher odds of achieving high physical activity levels than those who answered yes (48.3% versus 21.4%; OR = 2.35; 95% CI, 1.19–4.63). In the stratified analyses, females who reported that none of their friends smoke had over 2.3 times higher odds of achieving high physical activity levels compared to the adolescents who reported that all or most of their friends smoke (47.6% versus 19.6%: OR = 2.38; 95% CI, 8.22–6.87).

These findings for weekday screen time, smoking behaviour and peer smoking and alcohol behaviours were also apparent in the series of models adjusted for sex, age and state site (Table 3). As well, compared to participants with high weekend screen time, those with low weekend screen time had 1.6 times higher odds of achieving high physical activity levels among the whole sample (54.3% versus 43.4%: OR = 1.61; 95% CI, 1.22–2.13) and for males (60.5% versus 48.3%: OR = 1.84; 95% CI, 1.22–2.79) but not females. Among the whole sample, participants who reported that none, a few or about half of their friends smoke had 2.2 times, 1.8 times and 2.4 times higher odds of achieving high physical activity levels, respectively, compared to those reporting that all or most of their friends smoke (50.4% versus 25.0%: OR = 2.24; 95% CI, 1.24–4.05; 39.8% versus 25.0%: OR = 1.80; 95% CI, 1.02–3.16; 46.2% versus 25.0%: OR = 2.36; 95% CI, 1.22–4.59, respectively). Among females but not males, those who reported that none or about half of their friends smoke had 2.2 times and 2.4 times higher odds of achieving high physical activity levels, respectively, compared to those reporting that all or most of their friends smoke (47.8% versus 19.6%: OR = 3.06; 95% CI, 1.35–6.92; 45.0% versus 19.6%: OR = 3.29; 95% CI, 1.33–8.17, respectively). Among the whole sample, those who had never tried smoking had over 1.8 times higher odds of achieving high physical activity levels than those who answered yes (50.8% versus 25.0%: OR = 1.86; 95% CI, 1.30–2.65). Among females but not males, participants who reported that about half of their friends drank alcohol had 2.7 times higher odds of achieving high physical activity levels compared to those reporting that all or most of their friends smoke (57.5% versus 50.0%: OR = 2.72; 95% CI, 1.29–5.75).

Participants who answered ‘don’t remember’ to the physical activity question had higher odds of answering ‘don’t remember’ to the weekday recreational screen time (OR = 6.76; 95% CI, 2.70–16.77) in the fully adjusted model (Table 2). There were also associations between some other health and other movement behaviours and answering ‘don’t remember’ to the physical activity questions in the analysis adjusted for sociodemographic variables (Table 3). Participants responding ‘don’t remember’ regarding their physical activity also had higher odds of answering ‘don’t remember’ regarding their weekday (OR = 11.21; 95% CI, 6.74–18.64) and weekend (OR = 9.21; 95% CI, 5.59–15.21) recreational screen time and lower odds for how many of their friends drink alcohol (about half versus all or most OR = 0.32; 95% CI, 0.14–0.71). As well, participants responding ‘don’t remember’ regarding very good sleep quality, compared to fairly/very bad (OR = 1.95; 95% CI 1.12–3.42) or who had never tried a full serve of alcohol (OR = 2.08; 95% CI 1.11–3.91) had higher odds of achieving high physical activity levels.

## 4. Discussion

This study has identified new associations of physical activity and other sociodemographic, movement and health related behaviours among Aboriginal adolescents and young people in the Next Generation Youth Wellbeing study in Australia. The strongest associations, in the fully adjusted analysis that included all participants, were found between high physical activity levels and being male, having low weekday screen time levels, non-smoking friends and few friends who drank alcohol. Some associations differed by sex: associations between physical activity and age; physical activity and weekday recreational screen time were found for males whereas associations between physical activity and having ever smoked of having friends who smoked were found for females There were additional associations between high physical activity levels and low weekend screen time levels.

Known sociodemographic correlates of physical activity of children, adolescents and young people internationally include sex and age where younger children and males typically have high physical activity levels [30]. This is also consistent with several physical activity studies with young Aboriginal participants [8,10,11,12]. While we observed a decline in physical activity by age, age was not significantly associated with physical activity in the adjusted model that included all participants, but there were significant associations among males. These associations may have been confounded by smoking and alcohol variables that differed by age and sex but the linear decline by age highlights the need for engaging physical activity strategies and programs within supporting settings for Aboriginal adolescents and young people.

These findings add new knowledge of how some health and peer behaviours apparently co-exist among Aboriginal adolescents and young people. The co-occurrence of protective behaviours relating to (high) physical activity, (less) screen time, peer non-smoking and non-drinking, with some similarities to Australian and international adolescent and young adult behaviour studies [17,31]. Among younger Aboriginal children age 5–9 years, clustering of high sports participation with low screen time was clear [15] which is similar to the present findings among adolescents and young people, although for overall physical activity rather than sport participation specifically [4]. The relationships between physical activity and screen time among adolescents and young people can be partly due to one behaviour displacing or replacing the other in time use. Given their behavioural attributes and largely similar determinants and correlates, both can be targeted together [32]. Emerging evidence indicates that low levels of family screen time at age 0–5 predicts high physical activity among Aboriginal children at age 8–13 [33]. A deeper understanding of the nature of physical activity and screen time relationships among Aboriginal young people may assist in identifying future program strategies that may include family programs during early childhood. These programs may be particularly beneficial for males given the stronger associations found.

A previous analysis of NextGen data identified that more than half of participants did not smoke, drink alcohol or use marijuana with evidence of healthy clustering profiles relating to tobacco, alcohol and other drugs [34]. Physical activity can co-occur with both smoking and alcohol use among adolescents and young people [35] and this was evident in the present study for smoking, especially among females. The present study findings also show stronger associations between physical activity and peer behaviours where those whose friends did not smoke had high levels of physical activity, again especially among females. We also found that participants who reported about half of their friends drank alcohol, compared to all or most friends, had high physical activity levels although there were no associations with having fewer friends drink alcohol. These findings indicate that peer behaviours are an important part of the health behaviour associations of Aboriginal adolescents and young people that is demonstrated for the first time in this study and is consistent with the international literature [18,19]. The findings suggest healthy behaviours could be promoted through peer or role modelling alongside education and awareness initiatives that could be particularly beneficial for females.

In the present study, more (42%) participants achieved high levels of physical activity (at least 3 days per week) than the other two categories. This highlights the challenges in measuring physical activity among the NextGen cohort that could be attributed to the relevance of the measure or data collection procedures. While the physical activity question has been used in previous state level surveillance and studies with Aboriginal young people and self-reported by 12–17 year olds [23], providing examples, prompts and clarification to participants may assist in future data collection, as well as revisiting the relevance of the question with Aboriginal adolescents and young people. Participants who reported not remembering their recreational screen time had higher odds of also reporting not remembering their weekly physical activity participation. This suggests that increasing knowledge and awareness among adolescents and young people of the relevance and importance of these behaviours to health and wellbeing is required [6]. Increasing knowledge and awareness of healthy lifestyle behaviours and could be a part of broader strategies to increase physical activity levels and improve other behaviours associated with Aboriginal adolescent and young adult health and wellbeing [2]. Alternatively, participants may have been aware of the relevance and importance of being physically active and minimising screen time but that their time use did not reflect healthy criteria and preferred to answer that they did not remember. This is consistent with the partially adjusted analysis where participants who reported they did not remember their weekday or weekend recreational screen time had higher odds of reporting low levels of physical activity. Further sensitively conducted and culturally safe qualitative research may give a clearer understanding of Aboriginal adolescent and young adult views and perceptions and could help design supportive strategies to facilitate physical activity and reduce screen time. Such strategies could include Aboriginal community sport clubs and activities [36].

Strengths of the study include Aboriginal leadership, formative questionnaire development with Aboriginal adolescents and young people, data collection and interpretation of findings by Aboriginal research assistants. This assisted in the recruitment of a large sample of Aboriginal adolescents and young people from geographical diverse regions within three Australian states and territories. Participants were asked to consider their physical activity outside of school hours only; while this question would not have captured all physical activity of younger adolescents who attend school, the question used is consistent with a previous study with Aboriginal adolescents in Australia [12], and internationally with established reliability and validity [24]. Further, this question was suitable for the wide age range of participants in the present study, many of whom aged 10–24 years were older than school age. However, as 15% of participants answered that they did not remember how much physical activity they had conducted in the past week suggests that the relevance of the question as situated within existing data collection procedures could be improved, through meaningful collaboration with Aboriginal adolescents and young people. As well, the physical activity measure only includes the number of days per week; the total number of hours in a week spent doing physical activity may more accurately reflect participation. Objective devices to measure physical activity may have improved accuracy than subjective questionnaire measures but may also have additional limitations [37].

## 5. Conclusions

The NextGen study provides new evidence of associations between high physical activity levels and movement, healthy lifestyle and peer behaviours during adolescence, including recreational screen time, having never smoked, peer non-smoking and peer alcohol use. These findings inform ways to co-design and enhance holistic strategies to increase physical activity that are relevant to Aboriginal adolescents and young people that may include targeting peer behaviours and social connections and that may need to be specific for males and females. Follow-up data collection of the NextGen cohort is planned that will allow for question refinement and the examination of causal relationships between the associations of physical activity, movement, and healthy lifestyle and peer behaviours found in the present study.

## Figures and Tables

**Figure 1 ijerph-20-03738-f001:**
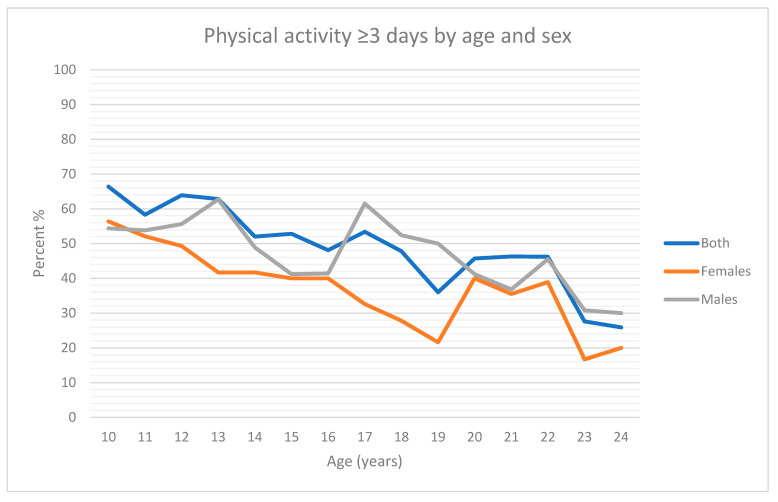
Percent of Next Generation Youth Well-being Study participants achieving ≥3 days a week of physical activity by age.

**Table 1 ijerph-20-03738-t001:** Participant characteristics for physical activity, sociodemographic and health variables among Next Generation Youth Well-being Study participants.

				High Physical Activity (≥3 Days/Week)		
			Yes	No	*p* Value *	Don’t Remember
	N	%	N (%)			
Total	1170	n/a	524 (41.9)	455 (36.4)		191 (15.3)
Sociodemographic						
Sex						
Female	661	57.3	266 (40.2)	274 (41.5)	0.002	121 (18.3)
Male	493	42.7	251 (50.9)	172 (34.9)		70 (14.2)
Age (years)						
10–11	265	22.6	146 (55.1)	85 (32.1)	<0.001	34 (12.8)
12–13	245	20.9	128 (52.2)	81 (33.1)		36 (14.7)
14–15	193	16.5	83 (43.0)	75 (38.9)		35 (18.1)
16–17	160	13.7	66 (41.3)	65 (40.6)		29 (18.1)
18–19	115	9.8	39 (33.9)	54 (47.0)		22 (19.1)
20–21	93	7.9	35 (37.6)	41 (44.1)		17 (18.3)
22–24	99	8.5	27 (27.3)	54 (54.5)		18 (18.2)
State site						
Central Australia	84	7.2	34 (40.5)	43 (51.2)	0.022	7 (8.3)
Western Australia	735	62.8	345 (46.9)	262 (35.6)		128 (17.4)
New South Wales	351	30.0	145 (41.3)	150 (42.7)		56 (16.0)
IRSEO tertiles						
Most advantaged	178	30.2	100 (44.1)	78 (34.4)	0.363	49 (21.6)
Middle advantaged	212	36.0	104 (41.6)	108 (43.2)		38 (15.2)
Most disadvantaged	199	33.8	102 (45.9)	78 (34.4)		23 (10.4)
Movement Behaviours						
Weekday recreational screen time						
High (≥3 h/day)	564	49.6	248 (44.0)	266 (47.2)	<0.001	50 (8.9)
Low (0–2 h/day)	434	38.1	240 (55.3)	137 (31.6)		57 (13.1)
Don’t remember	140	12.3	26 (18.6)	39 (27.9)		75 (53.6)
Weekend recreational screen time						
High (≥3 h/day)	569	49.9	247 (43.4)	263 (46.2)	<0.001	59 (10.4)
Low (0–2 h/day)	438	38.3	238 (54.3)	147 (33.6)		53 (12.1)
Don’t remember	138	12.1	27 (19.6)	38 (27.5)		73 (52.9)
Sleep quality						
Very good	430	38.3	204 (47.4)	152 (35.3)	0.02	74 (17.2)
Fairly good	517	46.1	230 (44.5)	203 (39.3)		84 (16.2)
Fairly bad/very bad	175	15.6	66 (37.7)	85 (48.6)		24 (13.7)
Health behaviours						
Ever tried smoking						
No	799	70.4	406 (50.8)	274 (34.3)	<0.001	119 (14.9)
Yes	302	26.6	94 (31.1)	154 (51.0)		54 (17.9)
Prefer not to say	34	3.0	9 (26.5)	14 (41.2)		11 (32.4)
How many friends smoke						
None	476	45.1	240 (50.4)	167 (35.1)	<0.001	69 (14.5)
A few	387	36.7	154 (39.8)	163 (42.1)		70 (18.1)
About half	104	9.9	48 (46.2)	43 (41.3)		13 (12.5)
All of most	88	8.3	22 (25.0)	50 (56.8)		16 (18.2)
Ever had a full serving of alcohol						
No	650	67.6	318 (48.9)	227 (34.9)	<0.001	105 (16.2)
Yes	312	32.4	119 (38.1)	153 (49.0)		40 (12.8)
How many friends drink alcohol?						
None	362	37.9	174 (48.1)	137 (37.8)	0.004	51 (14.1)
Don’t know	66	6.9	23 (34.8)	26 (39.4)		17 (25.8)
A few	255	26.7	99 (38.8)	110 (43.1)		46 (18.0)
About half	92	9.6	50 (54.3)	30 (32.6)		12 (13.0)
All or most	179	18.8	63 (35.2)	92 (51.4)		24 (13.4)

* Chi-squared tests of association; N of variable totals less than overall N due to small proportions of missing data.

**Table 2 ijerph-20-03738-t002:** Adjusted odds ratios (OR), and their 95% confidence intervals (95% CI) of NextGen participants achieving high physical activity (≥3 days: reference low physical activity 0–2 days), adjusted for sociodemographic, movement and health variables, among the whole sample and stratified by sex.

	All		Females		Males	
Variable	OR (95% CI)		OR (95% CI)		OR (95% CI)	
	High Physical Activity	Don’t Remember	High Physical Activity	Don’t Remember	High Physical Activity	Don’t Remember
Sociodemographic						
Gender (ref male)						
Female	0.57 (0.41–0.81)	1.42 (0.82–2.47)	-	-	-	-
Age (years; ref 22–24)						
10–11	2.10 (0.80–5.52)	0.33 (0.08–1.27)	0.83 (0.23–3.05)	0.28 (0.05–1.45)	6.83 (1.43–35.57)	0.20 (0.01–3.14)
12–13	2.18 (0.89–5.35)	0.51 (0.15–1.67)	1.23 (0.37–4.11)	0.77 (0.20–3.07)	4.80 (1.10–20.99)	0.06 (0.00–0.93)
14–15	1.87 (0.83–4.23)	0.73 (0.26–2.03)	1.86 (0.63–5.49)	0.72 (0.21–2.50)	2.45 (0.63–9.44)	0.64 (0.08–5.07)
16–17	1.48 (0.65–3.35)	0.58 (0.19–1.74)	1.05 (0.36–3.03)	0.70 (0.20–2.42)	2.26 (0.57–8.91)	0.26 (0.01–5.00)
18–19	1.34 (0.58–3.12)	0.82 (0.28–2.38)	0.52 (0.16–1.72)	0.71 (0.20–2.54)	4.59 (1.06–19.96)	2.62 (0.30–23.34)
20–21	1.84 (0.79–4.26)	0.95 (0.30–2.97)	1.92 (0.67–5.54)	0.92 (0.23–3.73)	1.64 (0.38–7.08)	2.03 (0.21–19.54)
State site (ref New South Wales)						
Central Australia	1.17 (0.89–2.33)	0.64 (0.20–2.13)	0.97 (0.35–2.72)	0.53 (0.09–3.03)	1.27 (0.52–3.11)	0.30 (0.04–2.18)
Western Australia	1.10 (0.77–1.57)	1.17 (0.68–1.99)	0.85 (0.51–1.40)	1.18 (0.61–2.30)	1.39 (0.81–2.41)	0.72 (0.24–2.17)
Movement behaviours						
Weekday recreational screen time (ref high ≥ 3 h/day)						
Low 0–2 h/day	1.70 (1.11–2.61)	1.86 (0.93–3.72)	1.35 (0.73–2.51)	1.80 (0.76–4.25)	2.20 (1.12–4.30)	2.44 (0.58–10.31)
Don’t remember	0.84 (0.35–2.04)	6.73 (2.70–16.77)	0.87 (0.27–2.74)	5.01 (1.71–14.72)	0.69 (0.15–3.10)	44.44 (4.40–494.98)
Weekend recreational screen time (ref high ≥ 3 h/day)						
Low 0–2 h/day	1.41 (0.92–2.15)	0.97 (0.49–1.91)	1.33 (0.73–2.43)	0.84 (0.36–1.97)	1.49 (0.77–2.88)	0.92 (0.23–3.73)
Don’t remember	0.77 (0.33–1.82)	2.20 (0.89–5.41)	0.62 (0.19–2.01)	2.33 (0.81–6.74)	1.00 (0.26–3.92)	0.99 (0.10–9.99)
Sleep quality (ref fairly/very bad)						
Very good	1.18 (0.73–1.91)	1.78 (0.84–3.77)	1.27 (0.66–2.43)	1.59 (0.66–3.81)	1.18 (0.53–2.61)	3.69 (0.62–22.10)
Fairly good	0.91 (0.53–1.56)	1.92 (0.83–4.40)	0.95 (0.44–2.05)	2.02 (0.75–5.45)	0.99 (0.42–2.32)	3.65 (0.54–24.74)
Health behaviours						
Ever tried smoking (ref yes)						
No	1.44 (0.89–2.33)	0.69 (0.34–1.39)	2.35 (1.19–4.63)	0.76 (0.33–1.75)	0.87 (0.41–1.84)	1.42 (0.35–5.83)
Prefer not to say	0.87 (0.27–2.82)	2.08 (0.50–8.67)	0.72 (1.17–4.52)	1.58 (0.24–10.47)	0.87 (0.16–4.67)	5.41 (0.47–62.12)
Ever tried a full serve of alcohol (ref yes)						
No	0.78 (0.44–1.36)	1.96 (0.91–4.24)	0.80 (0.35–1.80)	1.86 (0.71–4.89)	0.71 (0.30–1.64)	3.98 (0.92–17.24)
Peer behaviours						
How many friends smoke (ref all of most)						
None	2.12 (0.96–4.71)	1.42 (0.44–4.62)	2.38 (8.22–6.87)	2.58 (0.62–10.72)	1.54 (0.40–5.89)	0.07 (0.00–1.22)
A few	1.52 (0.77–3.00)	1.58 (0.59–4.26)	1.47 (0.58–3.71)	2.63 (0.78–8.88)	1.44 (0.52–4.59)	0.22 (0.02–2.22)
About half	1.92 (0.89–4.11)	1.12 (0.35–3.60)	2.48 (0.86–7.17)	1.26 (0.29–5.53)	1.31 (0.39–4.41)	0.51 (0.05–5.29)
How many friends drink alcohol (ref all or most)						
None	0.87 (0.41–1.85)	0.90 (0.30–2.65)	1.19 (0.42–3.39)	0.73 (0.20–2.73)	0.60 (0.18–2.06)	2.16 (0.16–28.71)
Don’t know	0.53 (0.22–1.30)	0.75 (0.23–2.44)	0.45 (0.11–1.81)	1.28 (0.32–5.14)	0.50 (0.14–1.82)	0.08 (0.00–2.28)
A few	0.88 (0.49–1.57)	0.97 (0.42–2.26)	0.69 (0.32–1.53)	0.82 (0.30–2.21)	0.87 (0.32–2.38)	1.64 (0.19–14.03)
About half	2.08 (1.04–4.15)	1.20 (0.41–3.55)	2.09 (0.77–5.62)	1.72 (0.46–6.46)	1.81 (0.62–5.26)	0.46 (0.05–4.67)

Multinomial logistic regression model, adjusted for all other column variables. Abbreviations: CI—confidence interval; OR—odds ratio.

**Table 3 ijerph-20-03738-t003:** Adjusted odds ratios (OR), and their 95% confidence intervals (95% CI) of NextGen participants achieving high physical activity (reference: low physical activity 0–2 days), adjusted for sex, age, state site among the whole sample and stratified by sex (adjusted for age, state site).

	All		Females		Males	
	OR (95% CI)		OR (95% CI)		OR (95% CI)	
	High Activity	Don’t Remember	High Activity	Don’t Remember	High Activity	Don’t Remember
Movement behaviours						
Weekday recreational screen time (ref high ≥ 3 h/day)						
Low 0–2 h/day	1.72 (1.29–2.28)	2.33 (1.50–3.62)	1.32 (0.89–1.94)	2.13 (1.19–3.82)	2.29 (1.49–2.48)	2.63 (1.31–5.29)
Don’t remember	0.73 (0.42–1.26)	11.21 (6.74–18.64)	0.67 (0.35–1.29)	9.21 (4.93–17.21)	0.89 (3.21–2.48)	18.03 (6.91–47.05)
Weekend recreational screen time (ref high ≥ 3 h/day)						
Low 0–2 h/day	1.61 (1.22–2.13)	1.63 (1.06–2.50)	1.43 (0.97–2.11)	1.55 (0.90–2.67)	1.84 (1.22–2.79)	1.10 (0.41–2.94)
Don’t remember	0.76 (0.44–1.31)	9.21 (5.59–15.21)	0.68 (0.35–1.33)	6.35 (3.46–11.66)	1.85 (0.91–3.79)	19.79 (7.62–51.39)
Sleep quality (ref fairly/very bad)						
Very good	1.33 (0.88–2.01)	1.95 (1.12–3.42)	1.61 (0.92–2.81)	1.79 (0.91–3.55)	1.13 (0.60–2.16)	3.21 (1.13–9.09)
Fairly good	1.30 (0.88–1.91)	1.51 (0.89–2.57)	1.31 (0.78–2.19)	1.29 (0.69–2.41)	1.36 (0.73–2.51)	2.31 (0.64–6.37)
Health behaviours						
Ever tried smoking (ref yes)						
No	1.86 (1.30–2.65)	1.22 (0.77–1.92)	3.14 (1.92–5.15)	1.47 (0.84–2.58)	0.97 (0.55–1.71)	0.98 (0.44–2.19)
Prefer not to say	0.84 (0.34–2.07)	2.01 (0.84–4.84)	1.12 (0.35–3.57)	1.28 (0.40–4.12)	0.62 (0.14–2.73)	3.68 (0.83–16.38)
Ever tried a full serve of alcohol (ref yes)						
No	0.96 (0.60–1.54)	2.08 (1.11–3.91)	1.19 (0.62–2.27)	2.19 (1.02–4.72)	0.74 (0.36–1.50)	1.90 (0.61–5.91)
Peer behaviours						
How many friends smoke (ref all of most)						
None	2.24 (1.24–4.05)	1.22 (0.61–2.46)	3.06 (1.35–6.92)	1.71 (0.68–4.31)	1.50 (0.59–3.80)	0.73 (0.22–2.40)
A few	1.80 (1.02–3.16)	1.21 (0.63–2.30)	1.88 (0.85–4.15)	1.75 (0.74–4.15)	1.70 (0.72–4.02)	0.71 (0.25–2.04)
About half	2.36 (1.22–4.59)	0.90 (0.38–2.10)	3.29 (1.33–8.17)	1.03 (0.33–3.27)	1.45 (0.52–4.01)	0.70 (0.18–2.65)
How many friends drink alcohol (ref all or most)						
None	1.31 (0.68–2.51)	0.75 (0.36–1.56)	1.32 (0.67–2.60)	1.79 (0.78–4.10)	0.70 (0.28–1.71)	4.51 (1.12–18.15)
Don’t know	1.35 (0.70–2.59)	0.58 (0.28–1.18)	1.01 (0.38–2.70)	2.70 (0.98–7.49)	0.54 (0.19–1.55)	3.53 (0.86–14.50)
A few	2.66 (1.24–5.69)	0.54 (0.21–1.36)	1.16 (0.64–2.10)	1.62 (0.80–3.28)	0.87 (0.41–1.84)	1.88 (0.60–5.84)
About half	2.22 (1.26–3.90)	0.32 (0.14–0.71)	2.72 (1.29–5.75)	1.42 (0.51–4.02)	1.30 (0.52–3.23)	1.81 (0.44–7.47)

Multinomial logistic regression models. Abbreviations: CI—confidence interval; OR—odds ratio.

## Data Availability

Data are available from the authors on reasonable request.

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
