# Peer review of "Social and Behavioural Correlates of High Physical Activity Levels among Aboriginal Adolescent Participants of the Next Generation: Youth Wellbeing Study"

_ijerph, 2023, doi:10.3390/ijerph20043738_

Round 1

Reviewer 1 Report

Review Report by Reviewer 1

Dear editor,

Thank you very much for giving me the opportunity for reviewing this paper. This paper examined the associations between physical activity levels and sociodemographic, movement and health variables of Aboriginal teenages. The results of this study showed that most of adolescents had high levels of physical activity. Also, recreational screen time and being female were associated with high PA. Generally to speak, this study is interesting and valuable for some reasons. First of all, it uses an almost large sample size. Second, the methodology is very good. Finally, the paper has been organized and written very well. Totally, I would suggest publication for this paper. I just would have some major and minor issues that are listed below.

Major issues

1) The age of the subjects of this research was chosen between 10 and 24 years and they were mentioned as teenagers in the introduction and research method. However, according to international definitions, the age of adolescence is considered up to 19 years. Therefore, there is a contradiction in this research with international definitions that needs to be explained by the authors.

2) Please divide the Method into several parts such as participants, instruments, etc.

3. Please mention validity and reliability of the instruments used in this study.

Minor issues

1) I understand that large sample size makes it almost impossible to use accelerometers for measuring PA, however, it should be mentioned as a limitation for this study as PA was measured subjectively and not objectively. 

2) Line 79-80, please add a reference for this claim.

3) Line 363, please correct "movement".

Good luck

Author Response

1. Thank you very much for giving me the opportunity for reviewing this paper. This paper examined the associations between physical activity levels and sociodemographic, movement and health variables of Aboriginal teenages. The results of this study showed that most of adolescents had high levels of physical activity. Also, recreational screen time and being female were associated with high PA. Generally to speak, this study is interesting and valuable for some reasons. First of all, it uses an almost large sample size. Second, the methodology is very good. Finally, the paper has been organized and written very well. Totally, I would suggest publication for this paper. I just would have some major and minor issues that are listed below.

Thank you for your helpful review and suggestions.

2. Major issues

The age of the subjects of this research was chosen between 10 and 24 years and they were mentioned as teenagers in the introduction and research method. However, according to international definitions, the age of adolescence is considered up to 19 years. Therefore, there is a contradiction in this research with international definitions that needs to be explained by the authors.

For the NextGen study, the age of focus is 10 – 24 years that is outlined in the study protocol (Gubhaju et al) but we acknowledge the international definition.

We have updated the second sentence to “One third of Aboriginal people are aged 10 -24 years, the period typically defined as adolescence and young adulthood” with an additional supporting reference and have updated the paper to use the term “adolescents and young people” throughout or replaced with the term “participants’.

Teenage is mentioned in the introduction in relation to the timing of physical activity declines in existing national data and previous studies. The term is not in the Method.

3 Please divide the Method into several parts such as participants, instruments, etc.

We have included subheadings 2.1 Participants, 2.2. Measures 2.3 Data analysis

4. Please mention validity and reliability of the instruments used in this study.

 We have updated lines 122 – 124 to “Survey questions were identified by investigators and sourced, where possible, from existing studies with Aboriginal people that used reliable and valid measures.” and referenced reliability and validity data on the physical activity measure (line 126) and references are already provided to reliability and validity papers on lines 149-154.

5. Minor issues

I understand that large sample size makes it almost impossible to use accelerometers for measuring PA, however, it should be mentioned as a limitation for this study as PA was measured subjectively and not objectively.

We agree and have added mention as a limitation “Objective devices to measure physical activity may have improved accuracy than subjective questionnaire measures but may also have additional limitations” (lines 383-385) with a supporting reference.

6. Line 79-80, please add a reference for this claim.

We have added the supporting reference from the explanation in paragraph 2.

7. Line 363, please correct "movement"

We have corrected this term

Reviewer 2 Report

1    Identifying the social and behavioral determinants of high physical activities among Aboriginal Adolescents is an important subject, and the authors' efforts are commendable.

2.      My first concern with this work is the improper definition of adolescent. The adolescent age is supposed to be between 10 and 19 years old. If the authors, choose to extend the adolescence age to 24 years, then a strong rationale for such decision should be provided. Otherwise, this study's title/aim and inclusion criteria are flawed by misclassifying 20-24 years old participants as adolescents. I suggest using adolescents/young adults instead of calling all participants adolescents.

3.      Introduction: While the introduction provided a good background for the need for physical activity, it needs a biomedical basis for exploring physical activity among aboriginals. Reports on higher rates of cardiometabolic disorders or obesity, which are usually consequences of physical inactivity among aborigines versus other demographics, will provide strong justification for the study.

4.      Terminologies such as pre-adolescence should not be used if not defined. In fact, the authors need to define the developmental age groups in the introduction (e.g., infants, toddlers, children, pre-adolescent, adolescent, young adults, etc.).

5.      It is difficult to understand the meaning of the sentence in line 72 because of the word "cluster". Cluster is misused in that sentence. Words like correlation, association, or relationship will be clearer.

6.      Please perform English check on lines 63-71. That paragraph needs to be clarified.

7.      Line 74 is unclear. The term health behavior means two things. It could mean good health behavior or bad health behavior. So, what type of health behavior clusters among Aboriginals?

8.      Why did you retain the "don't remember" category in the outcome variable? The focus of the study is participants with high versus low physical activity. I recommend you exclude the "don't remember" participants. That will give you 2 categories in the outcome variable, and you can perform a binary logistic regression.

9.      Did you try to analyze the outcome variable as a count data using poisson regression/negative binomial. This may provide a more robust estimate than categorizing the outcome.

10.  I'd like to know in what instance you used a generalized linear mixed model. Please clarify.

11.  Please provide footnotes for all the tables. These footnotes should state the statistical test used.

12.  Table 1: How can you generate a p-value for a multiple-by-multiple contingency tables? It is not statistically correct. You can only generate p values for multiple by 2 or 2 by 2 contingency tables. If you exclude the "don't remember" group from your study participants, you would be able to provide p values; otherwise, you will need to remove the p values

13. The discussion/conclusion might change if the aforementioned recommendations are adopted.

Author Response

1. Identifying the social and behavioral determinants of high physical activities among Aboriginal Adolescents is an important subject, and the authors' efforts are commendable.

 Thank you for your helpful review and suggestions.

2. My first concern with this work is the improper definition of adolescent. The adolescent age is supposed to be between 10 and 19 years old. If the authors, choose to extend the adolescence age to 24 years, then a strong rationale for such decision should be provided. Otherwise, this study's title/aim and inclusion criteria are flawed by misclassifying 20-24 years old participants as adolescents. I suggest using adolescents/young adults instead of calling all participants adolescents.

For the NextGen study, the age of focus is 10 – 24 years that is outlined in the study protocol (Gubhaju et al) and the Azzopardi et al paper but we acknowledge the international definition.

We have updated the second sentence to “One third of Aboriginal people are aged 10 -24 years, the period typically defined as adolescence and young adulthood” with an additional supporting reference and have updated the paper to use the term “adolescents and young people” throughout, or replaced the term “participants’.

3. Introduction: While the introduction provided a good background for the need for physical activity, it needs a biomedical basis for exploring physical activity among aboriginals. Reports on higher rates of cardiometabolic disorders or obesity, which are usually consequences of physical inactivity among aborigines versus other demographics, will provide strong justification for the study.

Viewing Aboriginal health solely through a biomedical basis without acknowledging more complex and holistic concepts has been criticised (e.g. Durey et al 2012  Reducing the health disparities of Indigenous Australians: time to change focus). However there are important studies that show relationships between physical activity and chronic disease among Aboriginal people that have now been described in the Introduction, first paragraph “Evidence from systematic reviews of studies with Aboriginal participants demonstrates the benefits of physical activity on key metabolic and health indicators such as reductions in weight and improvement in biomarkers including blood pressure and cholesterol”

4. Terminologies such as pre-adolescence should not be used if not defined. In fact, the authors need to define the developmental age groups in the introduction (e.g., infants, toddlers, children, pre-adolescent, adolescent, young adults, etc.).

We have removed the term pre-adolescence and specified the ages of the children in the particular study referred to. Our paper does not examine developmental ages across the life-course e.g. no infants or toddlers so we do not feel that all age group categories need to be defined, beyond the clarity to your point above to widen the description of the study age group to adolescents and young people.

5. It is difficult to understand the meaning of the sentence in line 72 because of the word "cluster". Cluster is misused in that sentence. Words like correlation, association, or relationship will be clearer.

We have removed the term cluster.

6. Please perform English check on lines 63-71. That paragraph needs to be clarified.

We have checked and clarified this paragraph.

7. Line 74 is unclear. The term health behavior means two things. It could mean good health behavior or bad health behavior. So, what type of health behavior clusters among Aboriginals?

We have clarified that “There is evidence that health behaviours cluster group together among Aboriginal adults where being physically active may group with non-smoking and low or no alcohol consumption”

8. Why did you retain the "don't remember" category in the outcome variable? The focus of the study is participants with high versus low physical activity. I recommend you exclude the "don't remember" participants. That will give you 2 categories in the outcome variable, and you can perform a binary logistic regression.

While our main interest was the factors associated with high physical activity levels, we wish to retain the “don’t remember” category because this answer option was selected by 15% of the sample, indicating as described in the discussion “However, as 15% of participants answered that they did not remember how much physical activity they had done in the past week suggests that the relevance of the question as situated within existing data collection procedures could be improved, through meaningful collaboration with Aboriginal adolescents and young people.” We believe that by highlighting this issue we can advocate for improved physical activity measurement among Aboriginal adolescents and young people. Further, there were several significant findings (lines 273 – 286) among the “don’t remember” group that are discussed in lines 350 – 360, with this group identified as an important target group for increasing knowledge and awareness of healthy behaviours. 

9. Did you try to analyze the outcome variable as a count data using poisson regression/negative binomial. This may provide a more robust estimate than categorizing the outcome.

The physical activity outcome variable categories were chosen as they are closely aligned to physical activity guidelines. We have added detail of this to the methods (lines 135-6)

10.  I'd like to know in what instance you used a generalized linear mixed model. Please clarify.

Apologies for the confusion, mention of these models has been updated to multinomial logistic regression

11.  Please provide footnotes for all the tables. These footnotes should state the statistical test used.

We have added footnotes for all the tables that state the statistical test used.

12.  Table 1: How can you generate a p-value for a multiple-by-multiple contingency tables? It is not statistically correct. You can only generate p values for multiple by 2 or 2 by 2 contingency tables. If you exclude the "don't remember" group from your study participants, you would be able to provide p values; otherwise, you will need to remove the p values

As we have explained above, we wish you retain the “don’t remember” group. However, we agree with your point and have updated the chi squared tests to include only the two physical activity categories (high physical activity; yes, no), rearranged the table columns to present the chi squared data together followed by the descriptive “don’t remember” data and updated section 2.3 data analysis and text accordingly. The p-value significance was unchanged for each of the variables in Table 1, with the exception of IRSEO tertiles. Accordingly, we removed the IRSEO tertile  supplementary analysis.

13. The discussion/conclusion might change if the aforementioned recommendations are adopted.

We have made minor amendments to the discussion and conclusion in response to your recommendations.

Reviewer 3 Report

This research is very meaningful. The method is also suitable. However, there are several points that need to be further explored: first, the selection basis of several measurement indicators proposed in the study, such as peer influence, smoking and other behaviors; The second is that the behavior of adolescents is greatly influenced by their parents and the family economic and cultural environment, which may also be an important factor to consider; Third, teenagers' education may affect their physical activities. Good luck

Author Response

This research is very meaningful. The method is also suitable. However, there are several points that need to be further explored: first, the selection basis of several measurement indicators proposed in the study, such as peer influence, smoking and other behaviors; The second is that the behavior of adolescents is greatly influenced by their parents and the family economic and cultural environment, which may also be an important factor to consider; Third, teenagers' education may affect their physical activities. Good luck

Thank you for your review and suggestions. We have added background from previous studies about how smoking and alcohol behaviours are influenced by peers and the family socio-economic (including education background) and cultural environment.

 “Further, adolescent and young adult smoking and alcohol behaviours are influenced by their peers as well as also by family, socioeconomic and cultural factors” (lines 80-81).

We have also reflected on the findings in relation to these studies in the Discussion “These findings indicate that peer behaviours are an important part of the health behaviour associations of Aboriginal adolescents and young people that is demonstrated for the first time in this study and is consistent with international literature. The findings suggest healthy behaviours could be promoted through peer or role modelling alongside education and awareness initiatives that could be particularly beneficial for females (lines 337-340).

Round 2

Reviewer 2 Report

Thank you for responding to my comments.

Reviewer 3 Report

The author revised and improved the article. The research contents, methods and conclusions of the article are effective. It can meet the standard of publication. congratulations